# The Use of Off-Label Medications in Newborn Infants Despite an Approved Alternative Being Available—Results of a National Survey

**DOI:** 10.3390/pharmacy10010019

**Published:** 2022-01-25

**Authors:** Alex Veldman, Eva Richter, Christian Hacker, Doris Fischer

**Affiliations:** 1Department of Pediatrics, Helios HSK, 65199 Wiesbaden, Germany; doris.fischer@helios-gesundheit.de; 2Department of Pediatrics, Justus-Liebig University, 35385 Giessen, Germany; eva-richter@gmx.eu; 3The Ritchie Centre, Hudson Institute of Medical Research, Monash University, Melbourne 3168, Australia; 4Pharmacy, St. Vincenz Hospital, 65549 Limburg, Germany; c.hacker@st-vincenz.de; 5Department of Pediatrics, J.W. Goethe University Hospital, 60590 Frankfurt, Germany

**Keywords:** off-label drug use, neonate, NICU, European pediatric regulation, prescribing

## Abstract

Neonates continue to be treated with off-label or unlicensed drugs while in hospital. However, some medications that have previously been used in adults underwent clinical testing and licensure for use with a different indication in the neonatal and pediatric population. Almost always, the marketing of these newly approved substances in a niche indication is accompanied by a steep increase in the price of the compound. We investigated the use of the approved formulation or the cheaper off-label alternative of Ibuprofen (Pedea^®^), Propanolol (Hemangiol^®^) and Caffeine Citrate (Peyona^®^) in neonatal clinical practice by conducting a National Survey of 214 Perinatal Centers in Germany. We also assessed price differences between on- and off-label alternatives and the extend of the clinical development program of the on-label medication in the neonatal population. On-label medication was more frequently used than the off-label alternative in all indications (PDA: on-label to off-label ratio 1:0.26, Apnea: 1:0.56, Hemangioma 1:0.76). All sponsors did conduct placebo-controlled Phase III trials with efficacy and safety endpoints in the target population and the number of participants in the target population varied between 82 and 497. Costs for the three drugs in their approved and marketed formulations increased in median 405-fold compared with the corresponding off-label alternative. Overall, about one out of three neonatologists prescribed an off-label or non-approved drug to patients despite an alternative medication that is approved for the indication in the target population being available.

## 1. Introduction

The robust and vigorous testing of the efficacy and safety of new drugs in the relevant population prior to approval by regulatory authorities and entry into market is regarded as one of the key achievements in safe prescription practice, yet, less than 50% of the drugs commonly prescribed in neonatal medicine have market approval for use in this age group or have ever been pre-clinically or clinically investigated for safety or efficacy in this population [1]. This results in about 80% of all patients admitted to neonatal intensive care units (NICUs) being treated with at least one off-label/unlicensed medication during their stay in hospital [2]. Despite US and European legislation having reacted to this situation by introducing the Pediatric Research Equity Act in the US (2003) and the EC Regulation 1901/2006 and 1902/2006 in the European legislation (2006) [3,4] the percentage of neonates being treated with off-label or non-approved drugs remains disappointingly high today and seems to be increasing rather than decreasing, at least in some areas [5,6,7].

Some manufactures, however, did indeed conduct development programs to investigate the efficacy and safety of already-established medications, now to be used for a novel indication in the neonatal population—not surprisingly, accompanied by a significant price increase for the newly approved “old” drug. For example, intravenous Ibuprofen, used in neonates to close a hemodynamic relevant persistent ductus arteriosus (PDA), is currently marketed in two different preparations: one is Ibuprofen-lysine (Pedea^®^, Orphan Europe, Paris) the other is ibuprofen-arginine (Caldolor^®^, Cumberland Pharmaceuticals, Nashville, TN, USA). Ibuprofen-arginine, at a cost of approximately 16 € per 400 mg ampoule (0.04 €/mg), is an analgesic for adult patients with a statement in the product information sheet that safety and effectiveness has not been established in patients less than 17 years of age. Ibuprofen-lysine, in contrast, has market approval for the treatment of PDAs in preterm neonates and is available at a price of approximately 162 € per 10 mg (16.22 €/mg) vial, which corresponds to a 405-fold price increase. Other examples for treatments with a drastic price increase after licensure in the neonatal population are propanolol (Hemangiol^®^, Pierre Fabre Dermatologie, Lavaur, France) for the treatment of hemangioma and caffeine citrate (Peyona^®^, Chiesi Farmaceutici, Parma, Italy) for the treatment of neonatal apnea/bradycardia.

In this paper, we report the results of a national survey of all 214 perinatal centers in Germany, with the aim of assessing the use of either the approved expensive formulation of the above mentioned drugs or the cheaper off-label alternative in clinical practice. Additionally, we analyze the European Medicines Agency (EMA) public assessment reports (EPAR) on the clinical development programs of these drugs to understand to what extend additional clinical and non-clinical studies have been conducted by the respective sponsors to achieve market authorization in the neonatal indications.

## 2. Materials and Methods

### 2.1. Survey of Clinical Practice in German Perinatal Centers

A survey was mailed to all Level I (highest level of perinatal care, treating preterm neonates of all gestational age (GA) and birthweight) and Level II (admission of neonates with a birthweight of ≥1250 g and a GA above 28 weeks) neonatal centers in Germany (214 centers). Recipients were asked to specify the medication used to treat neonates with PDA (Pedea^®^ or others), neonates with apnea of prematurity (Peyona^®^ or others), neonates with hemangiomas (Hemangiol^®^ or others) and neonates with pulmonary hypertension (iNOmax^®^ or others, data not shown since nitric oxide (NO), as a technical gas, is no longer available for hospitals). Additionally, centers were asked to specify how many patients with each condition were treated per site per annum. Non-responders were followed-up with a phone call, and, if in agreement, submission of a second copy of the questionnaire. Responses could be returned anonymously. A copy of the survey is available in Appendix A.

### 2.2. Analysis of EMA’s EPARs

The Scientific discussion of the Public Assessment Reports were analyzed to understand the non-clinical and clinical studies that have been initiated and conducted by the respective sponsor/Marketing Authorization Application (MAA) holder. Clinical studies were then investigated to understand the study design (placebo controlled vs. non-controlled studies vs. compassionate use programs) number of subjects per study as well as overall number of participants in the target population (neonates) in the clinical development program. Post-marketing commitments were extracted, if any.

### 2.3. Drug Price Comparison

Prices for the three drugs in their approved and marketed formulation and corresponding off-label alternative have been sourced from a large hospital pharmacy serving a group of 18 hospitals in Germany (St. Vincenz Apotheke, Limburg, Germany).

### 2.4. Ethical Review Board (ERB) Approval

Neither ERB approval nor patient written informed consent was required because we used either publicly available data or de-identified statistical information (in the survey). No patient medical records were submitted or used.

## 3. Results

### 3.1. Survey of Clinical Practice in Perinatal Centers in Germany

Of the 214 Perinatal Centers, 151 (71%) responded. The results of the questionnaire for all three medications are summarized in Figure 1. Briefly; the on-label medication was more frequently used than the off-label alternative in all indications (on-label to off-label ratio 1:0.26 for the treatment of PDA with Pedea^®^, 1:0.56 for the treatment of apnea with Peyona^®^ and 1:0.76 for the treatment of Hemangiomas with Hemangiol^®^). Of note, the on-label group did include centers that use the on-label medication occasionally amongst other (off-label) treatments, if these were excluded, the ratio would naturally have shifted more towards the off-label use (see Figure 1).

The majority of centers conducted between 0–50 treatments in each indication per year with the exception of apnea of prematurity, in which the most centers conducted 50–100 treatments per year (further details in Figure 2).

### 3.2. Analysis of EMA’s EPARs

The number of subjects enrolled in clinical trials to achieve market approval in the neonatal indication varied greatly between the three compounds (Table 1). Investment in human Phase I studies pharmacokinetic (PK) studies in healthy volunteers was limited in all development programs (0–18 participants). All sponsors conducted placebo controlled, double blind Phase III trials with efficacy and safety endpoints in the target population. Tthe number of participants in the target population varied between 82 and 497.

Two out of the three compounds conducted a dose finding/PK study in the target population and all three manufactures committed to some form of post-marketing safety surveillance studies or registries (Table 2).

### 3.3. Drug Price Comparison

Costs for the three drugs in their approved and marketed formulation increased in median 405-fold compared with the corresponding off-label alternative, with the largest increase noted in caffeine for the treatment of apnea (see Table 3). Interestingly, the medication with the steepest relative price increase involved the lowest number of trial subjects in the clinical development program (Peyona^®^ for the treatment of neonatal apnea).

## 4. Discussion

The extent of scientific evidence that is required to establish that a new therapeutic agent has benefits that outweigh its risks remains an area of critical debate amongst clinicians and regulators alike. In this context, it is essential to differentiate between off-label use (using an approved drug in an indication, population or dosage other than stated in the label) versus unlicensed drugs (using a drug or chemical compound that has not been approved for any use in humans) versus compassionate-use (using a drug that is currently investigated in clinical studies in a patient that would not be eligible for the study due to not meeting inclusion/exclusion criteria). Two of the drugs investigated in this study have been converted from off-label to on-label use in neonates (intravenous ibuprofen for the treatment of PDA and propranolol for the treatment of hemangiomas), while the other has previously been used in the form of a magisterial preparation (caffeine for apnea of prematurity), and as such, has converted from unlicensed to approved.

We investigated the uptake of drugs that have been studied and approved for the use in neonates and newborns by a group of concerned clinicians who have lamented the paucity of data on efficacy and safety when treating their small and vulnerable patients with off-label medications for decades. While the numbers of centers using the approved formulation outweighed those that treated patients with off-label or unlicensed drugs, about one out of three neonatologists/pediatricians seems to be comfortable in prescribing an off-label or unlicensed drug to patients despite an approved alternative being available. One factor in this reluctant uptake might be that most neonatologists are used to prescribing medications outside the labeled indication, dosage or population, indicating some necessity for the acceptance of off-label use. A more cynical explanation would be that once efficacy and safety have been established, some cost-conscious prescribers would gladly revert to the much cheaper alternative compound, particularly since the increase in costs was as great as 1700-fold. However, the most extensive off-label use was documented in the drug with the smallest difference in costs between on- and off-label alternative, refuting economic burden as the main driver in decision-making in the centers surveyed in this study. This may be different in other geographic locations, where financial constrains might limit access to more expensive medications for many clinicians.

It is worthwhile to remember that from the business perspective of a pharmaceutical company, all three indications are small markets and rare enough to fulfill orphan drug criteria (prevalence of 5 or less people in 10,000 in the EU) for two out of three indications (PDA: 2.13/10,000 and apnea of prematurity: 0.5–1.2/10,000) [12,13]. Medical treatment of PDA was described in 4202 infants during a 4-year period in South Korea (population of 51.2 million) by Park et al. in an analysis of national epidemiologic data [14]. Caffeine is one of the top five most prescribed treatments in neonatology, 156 exposures, 199 courses and 3908 days of use per 1000 infants discharged from 305 NICUs managed by the Pediatrix Medical group from 2005–2010 (450,386 infants) [15]. Anderson et al. described the incidence of infantile hemangioma, the third indication, with 1.64 per 100 person years in a predominantly white, non-Hispanic US population. However, treatment was only required in 8–9% of cases [16]. Overall, the socio-economic and hospital budgetary impact of price changes for all three medications is likely to be relatively low in most developed countries.

Interestingly, there was no difference in the uptake of the drug that had not even been approved in any indication compared to those that had previously been used extensively in a different indication. However, independently of the prescriber behavior, one could question how much, if any at all, new knowledge was generated in the approval process for the neonatal indication. In this context, it is important to acknowledge that off-label is not the same as off-knowledge, and substantial evidence is generated by large academic studies outside of industry-sponsored Phase 3 trails that, therefore, might never be considered in a label. In the clinical development programs analyzed here, dose range and PK studies in the target population had been conducted in 2/3 of the compounds and safety-focused post-marketing commitments (in addition to standard Periodic Safety Update reports (PSURs) that are mandatory for all drugs) have been imposed to 2/3 of new drugs (Table 3). In the Summary of Product Characteristics (SPCs), population specific dosing instructions, the tabulated nature and the frequencies of adverse reactions and specific warnings are being made available to the prescriber, indicating a current and forthcoming gain in medical knowledge on the use of these medications in neonates that would not be available without the clinical development programs conducted for the purpose of licensure.

Obviously, this study has specific strengths and limitations. While the relatively high response rate of 71% points towards valid and representative results, the survey was restricted to a single country, Germany, and the results may not necessarily be generalized to other regions. While physicians practicing in Germany face a comparatively low risk of litigation (which, therefore, might increase the willingness to use an off-label alternative), there is no direct financial incentive for the prescribing physician per-se to use a cheaper drug. Nevertheless, since German hospitals’ remuneration is diagnosis-related groups (DRG)-based following a fixed case mix index (CMI) weight per case, lowering drug spending may play a role in strategic decisions on overall prescribing practice at an institutional level.

Finally, the lingering question on the appropriateness of the steep increase in the price of each compound remains. It is not in the scope of this paper to assess the costs of the clinical development program and therefore assign a price tag to the gain in medical knowledge achieved by licensure of these compounds for the respective neonatal indications. In 2018, Sinha et al. conducted an excellent study on 141 pediatric industry sponsored trials leading to 29 extended and 3 new indications, as well as new safety information for 16 drugs. The authors found that the median cost of investment for trials was 36.4 million $ (IQR, 16.6 to 100.6 million $) while among the 48 drugs with available financial information, median net return was 176.0 million $ (IQR, 47.0 to 404.1 million $), with a median ratio of net return to cost of investment of 680% (IQR, 80% to 1270%) [17]. Sinha et al. concluded that while these studies did provide important information about the effectiveness and safety of drugs used in children, costs to consumers have been high, and policymakers may consider directly funding such studies. In this context, it is worth noticing that using an off-label drug when other options are too expensive or not reimbursable by insurance companies is supported by a recent Joint Policy Statement by the European Academy of Pediatrics and the European Society for Developmental Perinatal and Pediatric Pharmacology [18]. Regardless of the moral implication of extreme price hikes, which have debated elsewhere [19,20,21] we believe that the uptake of new formulations of old drugs after these have been tested and approved in the neonatal population by pediatricians and neonatologists is the only way to credibly substantiate our demand for population specific safety and efficacy data for this vulnerable group of patients.

## Figures and Tables

**Figure 1 pharmacy-10-00019-f001:**
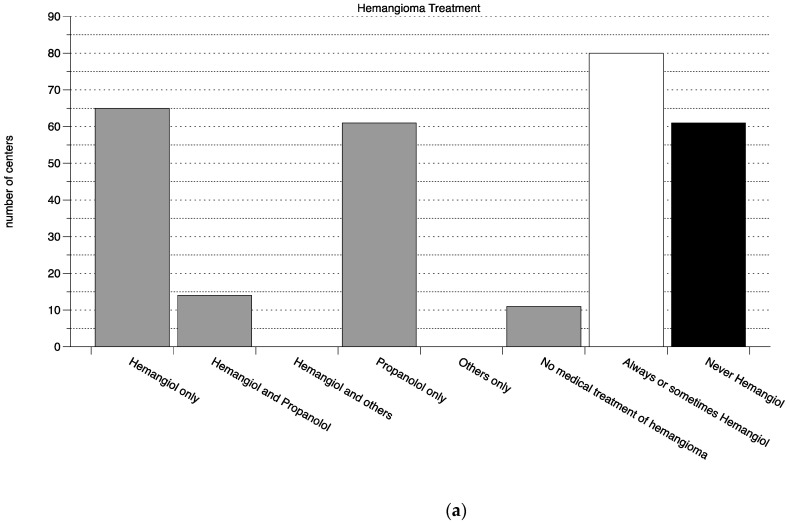
Use of on- and off-label medications. (**a**) Treatment of hemangiomas, (**b**) treatment of neonatal apnea, (**c**) treatment of PDA. From left to right, and for each indication in a separate panel, the number of centers is shown that use only the on-label medication, the on- and the off-label medication, the on-label and other medications, only the off-label, only other and no treatment (all bars shown in grey). On the right of each panel, the number of centers that always or sometimes treat with the on-label medication is shown in white and the number that never uses the on-label medication is shown in black. PDA—persistent ductus arteriosus.

**Figure 2 pharmacy-10-00019-f002:**
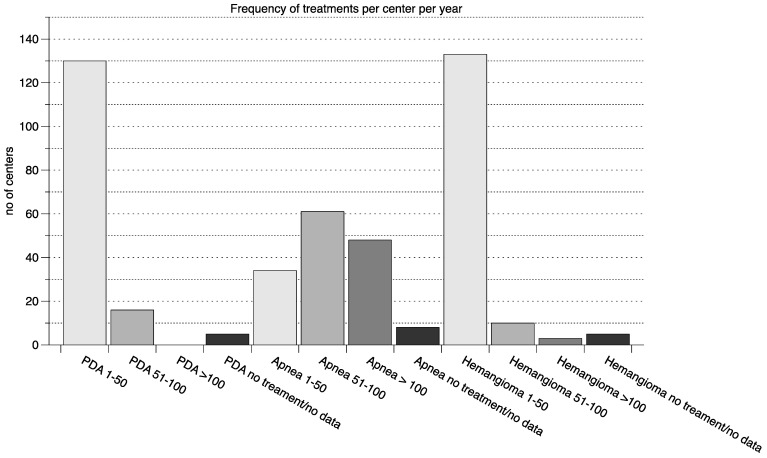
Frequency of treatments per center per year. The number of yearly treatments for the three indications (PDA, apnea and hemangioma) grouped in 1–50 treatments per year, 51–100 treatments per year and >100 treatments per year are shown (n of centers on the y axis). Additionally, for each indication the number of centers that do not use any drugs to treat in the indication or submitted no data is displayed. PDA—persistent ductus arteriosus.

**Table 1 pharmacy-10-00019-t001:** EPAR Analysis: Initial Development Program leading to Marketing Authorization.

Indication and Product	MAH Sponsored Studies		
	Non-Clinical	Clinical	
		Healthy Volunteers	Target Population
PDA product: Pedea^®^	iv toxicity study in in weaned and adult rats [8] (p. 4) local tolerance study in rabbits [8] (p. 4)	bioavailability study in 18 male healthy volunteers [8] (p. 16)	PK study in 62 preterm neonates [8] (p. 17) dose range study in 21 preterm neonates [8] (p. 23) double blind, placebo controlled trial in preterm neonates exposed to drug product (DP) n = 66; exposed to placebo n = 65 [8] (p. 24)
apnea of prematurity product: Peyona^®^	0	0	* double blind, placebo controlled trial in preterm neonates, exposed to DP n = 45; exposed to placebo n = 37 [9] (p. 16), [10]
hemangioma product: Hemangiol^®^	oral toxicity study in juvenile rats [11] (p. 24)	bioavailability/PK study in 12 male healthy volunteers [11] (p. 27)	open-label, repeated dose PK Study in 23 infants [11] (p. 27) double blind, placebo-controlled trial in infants exposed to DP n = 401; exposed to placebo n = 55 [11] (p. 28) open-label extension study (participants from previous studies, ongoing at the time of submission) [11] (p. 29) compassionate use program with 922 infants and children [11] (p. 54)

* Study conducted with a different drug product (DP, Cafcit^®^) for Food and Drug Administration (FDA) approval by a different MAH, EPAR: European Public Assessment Report, MAH—marketing authorization holder; iv—intravenous; PK—pharmacokinetics; PDA—persistent ductus arteriosus.

**Table 2 pharmacy-10-00019-t002:** EPAR Analysis: Studies performed Post Marketing Authorization.

Indication and Product	
PDA product: Pedea^®^	No studies conducted, label changes to PSUR analysis (gastric perforation added as risk)
apnea of prematurity product: Peyona^®^	European non-interventional post-authorization study to assess drug utilization and safety of caffeine citrate in the treatment of premature infants affected by apnea
hemangioma product: Hemangiol^®^	Updated efficacy and safety report of the pivotal study as well as the results of a small study conducted in France

EPAR—European Public Assessment Report; PSUR—periodic safety update reports; PDA—persistent ductus arteriosus.

**Table 3 pharmacy-10-00019-t003:** Price Comparison.

Indication	Product Approved in the Indication/Price in €	Alternative Product, Not Approved in the Indication/Price in €	Price Increase Not Approved to Approved in the Indication
PDA	Pedea^®^ 16.22 €/mg	Caldolor^®^ 0.04 €/mg	×405
apnea of prematurity	Peyona^®^ 0.86 €/mg	caffeine citrate 0.0005 €/mg	×1720
hemangioma	Hemangiol^®^ 0.56 €/mg	propanolol 0.015 €/mg	×37

PDA: persistent ductus arteriosus.

## Data Availability

Source data is available from the corresponding author on reasonable request.

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
