# Peer review of "The Use of Off-Label Medications in Newborn Infants Despite an Approved Alternative Being Available—Results of a National Survey"

_pharmacy, 2022, doi:10.3390/pharmacy10010019_

Round 1
Reviewer 1 Report
This study addressed an interesting and eternal problem for drug development and drug use in children, in particularly in neonates.
The paper is well written and easy to understand. If the design were an observational study on real administration of these 3 drugs in neonate centers, this study would add more value on its results because the survey reflected clinicians’ opinion rather than their practice. The practice would be changed according to local guidelines and drug availability
I have some minor comments:
- The questionnaire must be provided
- A brief discussion about the frequency of these three drugs use in neonate should be useful to discuss the benefit of sponsor on expensive price for approved formulations
- Line 184-185: “However, the most extensive off-label use was documented in the 184 drug with the smallest difference in costs between on- and off-label alternative, refuting 185 economic burden as the main driver in decision-making.” I did not agree with this statement; please provide reference for this statement. For example, in our NICU, the Hemangiol and Peyona are not available in our hospital because of economic choice. Many developing countries have to use other formulations of ibuprofen and paracetamol because they could not stand for the price of Pedea to treat PDA
Reviewer 2 Report
In the manuscript # pharmacy-1531539, by Alex Veldman and colleagues, “The use of off-label medications in newborn infants despite an approved alternative being available – results of a national survey.” authors investigated the use of the three approved drugs and the cheaper off-label alternatives of Ibuprofen, Propanolol and Caffeine Citrate in neonatal clinical practice by conducting a survey of 214 perinatal centres in Germany.
Authors report the results of a national survey focusing on a frequency of using approved formulations and off-label alternatives. Additionally, the status of clinical development programs for these off-label drugs is investigated.
This is a potentially interesting study, however in it requires some substantial improvements:
- Data about both clinical and non-clinical trials presented in the table 1 lacks references. It is crucial to provide them. Additionally, the reviewer suggests to “bullet-point” each of the study, so they are better separated in the text.
- The information about how many patients with each condition per year is treated (in total) is missing. Figure 2 presents frequency of treatments per centre per year, but it does not fully show the scale of discussed problem. Having this number would enable the readers, for example, to estimate and compare total costs of using approved medications and their off-label alternatives.
- Did the survey include questions about the reason of using cheaper off-label drugs? Authors provided short discussion about this issue, but maybe it can be extended by including some statistics or reasons/opinions of neatologists who agreed to take part in the survey.
- Authors should improve the graphs, especially Fig 1 a-c, which in the current form are non-readable. Increasing font size will make them easier to read.
- In the figure 2 legend, frequency intervals 0-50 should be changed to 1-50. 0 is already included in “no treatment/no data”.
